# Cost-Efficient Approaches for Fulfillment of Functional Coverage during Verification of Digital Designs

**DOI:** 10.3390/mi13050691

**Published:** 2022-04-28

**Authors:** Alexandru Dinu, Gabriel Mihail Danciu, Petre Lucian Ogrutan

**Affiliations:** 1Department of Electronics and Computers, Transilvania University of Brașov, 500036 Brașov, Romania; gabriel.danciu@unitbv.ro (G.M.D.); petre.ogrutan@unitbv.ro (P.L.O.); 2Technology, Siemens SRL, 500097 Brașov, Romania

**Keywords:** functional verification, hardware manufacturing, genetic algorithms, automation, data correlation, AI-enhanced processing, NSGA-II, SPEA2

## Abstract

Digital integrated circuits play an important role in the development of new information technologies and support Industry 4.0 from a hardware point of view. There is great pressure on electronics companies to reduce the time-to-market for product development as much as possible. The most time-consuming stage in hardware development is functional verification. As a result, many industry and academic stakeholders are investing in automating this crucial step in electronics production. The present work aims to automate the functional verification process by means of genetic algorithms that are used for generating the relevant input stimuli for full simulation of digital design behavior. Two important aspects are pursued throughout the current work: the implementation of genetic algorithms must be time-worthy compared to the application of the classical constrained-driven generation and the verification process must be implemented using tools accessible to a wide range of practitioners. It is demonstrated that for complex designs, functional verification powered by the use of genetic algorithms can go beyond the classical method of performing verification, which is based on constrained-random stimulus generation. The currently proposed methods were able to generate several sets of highly performing stimuli compared to the constraint-random stimulus generation method, in a ratio ranging from 57:1 to 205:1. The performance of the proposed approaches is comparable to that of the well-known NSGA-II and SPEA2 algorithms.

## 1. Introduction

The digital design of an electronic circuit is divided into two main parts: front-end development and back-end development. Front-end development involves several steps: realization of product specification, product design (usually performed using a Register Transfer Logic (RTL) language), functional verification, synthesis, and post-synthesis verification (also known as gate-level verification). Additionally, as a complementary process to verification, validation plays an important role in assessing the feasibility of functional requirements before the integrated circuit is manufactured. One of the most common methods of performing validation is the use of Field Programmable Gate Array (FPGA) devices [1,2]. Among the mentioned processes, the most time-consuming activity is functional verification [3]. Therefore, if the verification time is reduced, the whole Integrated Circuits (IC) manufacturing process is significantly reduced. An emerging approach, frequently encountered in industry-led and academic initiatives in this field, is to accelerate the functional verification process by using artificial intelligence techniques [4,5,6].

Functional verification aims to simulate all situations/states that a Design Under Test (DUT) may encounter after manufacture during actual operation. For example, in view of the hybrid turbocharger described in [7], the simulation of the digital design of the Electronic Control Unit (ECU) that controls a voltage relay must take into account all operating conditions of the device. They can be characterized by several parameters, as the fluctuations in coolant temperature, engine load, engine speed, and so on. Each state can be characterized by several stimulus sequences received at the ECU inputs. It is therefore necessary to create stimulus sets that bring the ECU into all its operating conditions in order to assess the correctness of its functionalities. Coverage points are used to record all possible operational situations. A coverage point is a specific verification structure that encodes different aspects of DUT operation. Each coverage element focuses on a specific functionality of the DUT extracted from the functional specification.

In addition, coverage collection mechanisms can be used to check the health of both the DUT and the verification environment. In the current work, a problem was detected because, during the verification of a module, the 100% coverage value was never reached for a given target. Examining the situation, an error in the reference model was found and fixed.

For example, a coverage point could consist of all states of a Finite State Machine (FSM). While running several simulations, the FSM reaches different states (in the current example, the number of states equals 4) and the coverage value increases (e.g., 25%, 50%, 75%, 100%) until it becomes 100% (the FSM has addressed all states in at least one simulation). In Figure 1, a possible state accessing sequence for FSM during the execution of 10 simulations is depicted. In this case, it is assumed that during one simulation, the DUT can only reach one state. The coverage value increases each time the FSM approaches a new state. The situation can become even more complex in the case of FSMs, because in addition to state coverage, transition coverage is used “to target faults in finite state machines (FSM)” [8]. However, this analysis is not the subject of this paper.

The verification process ends when all functional requirements are fully covered during the simulations performed. Therefore, the appropriate stimuli must be sent to the DUT inputs to reach all the bins of each coverage point. If the effect of the stimuli is redundant, the same bins will be addressed, and the coverage will not reach 100%. The classical solution to this problem is to generate random combinations of values for the DUT stimuli. The generation is constrained by the requirements found in the functional specifications related to the operation of the module. This type of approach is known as “constrained random generation” and is currently classically applied for design verification. It often happens that combinations of random input stimuli do not cause the DUT to reach all the bins of some coverage points, even after performing many simulations. In this case, verification engineers try to find out what stimuli values should be required to reach the coverage holes. The amount of work conducted by verification engineers is proportional to the complexity of the DUT. Therefore, for huge designs, finding the right stimuli to hit corner cases remains a challenge and a very time-consuming action. One very beneficial thing is to develop a method that incorporates “the DUT’s cause-effect relationships between the input stimuli sequences and the associated coverage results” [9]. One method to achieve this goal is to develop input stimulus sequences using genetic algorithms. These algorithms can allow sequences to over several iterations, until high-performing groups of inputs are generated, as shown in this paper. Genetic algorithms, used in the current work, are considered one of the best methods to implement the concept of Coverage Driven Test Generation (CDTG), which focuses on closing the feedback loop between the coverage score obtained by a simulation and the generation of input stimuli to further increase this score [4].

The verification process is accomplished by using several professional Electronic Design Automation (EDA) verification tools supplied by world’s leading Computer Aided Design (CAD) tool suppliers. However, these systems are not always affordable for small design centers and academia. Therefore, the development of efficient open-source or low-budget verification tools for digital ICs may be of interest to the potential customers mentioned above. In this paper, it is proposed to use Modelsim^®^ simulator, which is available free of charge under certain conditions.

To conclude this chapter, the object of our study is the functional verification of integrated circuits and the subject of our study is the automation of functional verification achieved by means of genetic algorithms.

Five main objectives are at the core of the current study:To provide readers with a straightforward but efficient automated mechanism for DUT input stimuli generation sequences; the implementation of this mechanism should be performed in a common high-level language that can interact with Electronic Design Automation (EDA) tools;To compare the currently developed approaches with two well-known evolutionary algorithms: NSGA-II and SPEA2;To highlight an original way of configuring genetic algorithms to achieve a better performance in terms of coverage fulfillment;To reduce the time needed for the functional verification process;To describe how a freely available tool (some limitations occur) can be successfully used in the functional verification process.

In order to prove the achievement of the above-mentioned goals, a DUT whose operation can be mapped onto an FSM is developed. A Python ecosystem, used to control the ModelSim^®^ simulator and to close the loop between coverage scoring and stimulus generation is created. Several simulations are run in different contexts to highlight the difference between currently developed approaches and constrained-random simulations.

The paper is organized as follows. Section 2 discusses some of the initiatives in industry and academia relevant to the topic of this paper and highlights the differences between their content and the present work. Section 3 provides background information on genetic algorithms and offers specific details about the approaches developed in this paper. This section also provides the necessary information on how to use the ModelSim^®^ simulator for advanced functional verification tasks. It presents the software environment created to control the hardware simulations and to process the results obtained and describes the design used to validate the benefits of the proposed approaches. Section 4 contains the results obtained belong with their description. The results, which are grouped considering the two analyzed coverage targets, are further interpreted in Section 5. Finally, Section 6 summarizes the achievements of this study.

## 2. Literature Review

The task of verification is a Nondeterministic Polynomial time (NP)-hard problem because of the high complexity of today’s digital ICs. To reduce the complexity of this task, heuristics based on machine learning, genetic algorithms, bio-inspired approaches, etc., are proposed in many industry and academic initiatives. The trade-off between minimum functional verification time and maximum coverage is the key point of all proposed techniques.

In a two-year study conducted by the authors, it was found that over 50% of approaches related to automating functional verification using artificial intelligence focus on increasing the level of coverage fulfillment.

Coverage performance can be achieved in a shorter time if correlations between input stimuli and associated coverage results are detected. According to [9], after observing the mentioned correlations, they can be used to train a model which is able to mimic the inverse function of the DUT: it receives as input the desired coverage outcome and returns the stimuli needed to achieve this coverage value. In addition, this model can be used to consequently generate the sets of stimuli that reach each component of the target coverage point. The approach developed in [10] uses supervised learning and requires a considerable amount of data for a well-trained model that subsequently predicts the stimuli needed to reach a given coverage bin. The present work, using genetic algorithms, creates the possibility to obtain high-performing sequences without the need of training a model. The large amount of training data required in [10] is replaced in the present work by running a reduced number of simulations during each iteration of the algorithm used, and the best-performing sequences obtained in each iteration are subsequently combined to obtain even better results in subsequent generations.

Research based on the existing literature shows that many initiatives that aim to automatically increase coverage fulfillment during functional verification process are related to mechanisms incorporating genetic algorithms. For example, in [11], the fitness function of genetic algorithms is represented by some elements of the DUT coverage fulfillment criteria. Therefore, sequence generation can be performed even before the DUT code was finished. In the current work, genetic algorithms need simulations performed on the actual digital design to evaluate the quality of a sequence. Therefore, the current approach is applicable even in the most complex cases where DUT behavior and coverage goals cannot be easily reflected in a software representation of the digital circuit.

The authors of [12] present a way to automate coverage fulfillment in Finite State Machines (FSM). A machine learning model is trained with the actions required to reach a state starting from another DUT state. Once the state-action-state network is created for the entire DUT, by analyzing randomly generated simulations, the shortest path to reach any DUT state becomes available, and 100% state coverage can be achieved using a small number of actions. Compared to [12], where the proposed methodology focuses only on state coverage metrics, this current study can provide good results for any type of coverage that can be properly represented.

The idea of linking stimulus sets to other elements of the verification environment has helped verification engineers to automate the verification process in other ways as well. Given the use of artificial intelligence in [13], links are created between transactions (test vectors) and triggered assertions (which are also automatically grouped into clusters). This helps to filter out transactions that do not contribute to coverage. Additionally, transaction generation can be influenced by a bias to obtain test vectors that better contribute to coverage fulfilment. In [13], supervised learning is used to analyze stimulus sequences. The best performing input groups are selected to be used in simulations. Compared to this situation, during the currently proposed versions of the developed algorithm, all generated sequences are simulated. Thus, the individuals (the structures used to manipulate the sequences during the genetic algorithm runs) that scored low are still sent to the crossover process because in some cases the resulting children might score unexpectedly high by reaching a corner situation in the DUT operation.

Another application of genetic algorithms in functional verification concerns the construction of input stimulus sequences that are effective in detecting various design errors [8]. In this case, after mutant versions of the DUT code are created, the sequences that succeed in detecting some of the mutations are further modified by the genetic operators to increase their error detection performance.

Coverage is the most important metric of the verification process and is intensely collected during each test run in the traditional approach [14]. However, collecting coverage makes simulations run slower. A solution to this problem is presented in [15], where most of the coverage data is predicted based on only a few collected pieces of information. Although this approach has been tested considering only “condition coverage” metric, [15] can be a starting point for developing new automation capabilities for functional coverage, too. The current work has in common with [15] the time savings achieved by performing fewer simulations compared to supervised learning- based methods.

Another approach related to coverage collection using data mining techniques is presented in [16]. The authors focused on automatically generating relevant coverage points for DUTs and testing whether they can be covered. Using this approach, the verification plan can be filled with situations that people can hardly think of, and hidden errors in corner case functionalities can be easily discovered. The approach developed here could be used to shorten the time needed to prove whether the coverage points generated in [16] are correct or infeasible.

Given the above review of some of the existing work in the literature, it can be shown that automating the verification process using coverage metrics is a hot topic, and the puzzle piece brought by the current work in this area is a welcome aid to support or inspire other efforts worldwide. 

## 3. Materials and Methods

### 3.1. Use of Genetic Algorithm Approaches

The current work focuses on automating coverage fulfillment using genetic algorithms. The operation of these algorithms is described in Section 3.1.1. The main outcome of the current work is the development of several genetic algorithm-based approaches that can be used or serve as models for automating stimulus generation in the functional verification process. These are presented in Section 3.1.2. To prove their effectiveness, and to emphasize that, in contrast to their ease of implementation, they perform unexpectedly well, the developed algorithms are compared with two well-established evolutionary algorithms that are presented in Section 3.1.3: Nondominated Sorting Genetic Algorithm II (NSGA II) [17] and Strength Pareto Evolutionary Algorithm 2 (SPEA2) [18].

#### 3.1.1. Genetic Algorithms

These algorithms are metaheuristic processes inspired by the theory of natural selection. The important aspect of this theory, also incorporated in genetic algorithms, is that offspring has similarities but also several differences compared to their parents. This aspect becomes more visible when considering a large number of generations of the same population, which continuously adapt to different environmental changes. Natural selection explains how beneficial variations are selected and passed on to subsequent generations. In contrast, harmful characteristics are poorly propagated in offspring, or even disappear completely. One of the simplest but most relevant examples is that, in the natural environment, unhealthy animals are the first to be caught by predators, and so are less likely to reproduce and pass on their disease. In this way, healthy and strong representatives of populations contribute most to the multiplication of species. Thanks to today’s powerful computing power, the evolution of generations can be simulated/calculated in seconds or less. Thus, genetic algorithms can provide solutions to optimization and search problems, drastically reducing operating time compared to brute force or chance-based alternatives.

In a genetic algorithm, the problem representation is structured so that the solution can be expressed by encoding it in an individual (also called chromosome) that “belongs” to a population. Each individual consists of several genes. The genes can be exchanged between individuals during the cross-over process or can be modified when mutation is performed.

The candidate solution “evolves” over several iterations/generations until the conditions for the optimal solution are met or a predefined number of iterations of the algorithm is reached.

Each evolution contains four stages that allow individuals to “mate” to eventually produce better offspring:Fitness calculation: this is how to calculate how close an individual’s representation is to the solution being sought.Selection: this stage is inspired by nature’s “survival of the fittest” paradigm, meaning that individuals whose representations are furthest from the optimal solution are simply discarded, keeping only those who might converge to the solution, as represented in Equation (1).
(1)selected_population={individual[i]:individual[i]∈CS, i<no_of_parents},
where *selected_population* represents the group of individuals which will play the parents’ role in the next population, *CS* (children_set) represents the individuals generated in the current population and *no_of_parents* represents the number of parents which must be selected from the children group at each new generation; in the current approach, this number is equal with the size of the initial population.Crossover: in this stage, an individual’s properties are shared and combined with other parts obtained from another individual of the same generation. The aim is to create new individuals whose fitness might be closer to the desired result. In the current study, the crossover process, in which two parents generated two children, followed the Equation (2).
(2)$$child_{1}=\{\begin{array}{c} parent_{1\;},\;if\;i\leq crossover\_point\\ parent_{2},\;if\;i>crossover\_point \end{array},\;child_{2}=\{\begin{array}{c} parent_{2\;},\;if\;i\leq crossover\_point\\ parent_{1},\;if\;i>crossover\_point\; \end{array},$$
where the *crossover_point* variable is a randomly generated number in range [1…no_of_genes−1].Mutation: this stage is inspired by the modifications that appear at individuals without being inherited from their parents. In biology, these mutations can be caused by environmental factors (e.g., UV or nuclear radiations), by internal reactions in the body which lead to DNA damage of some cells [19], etc. The aim of this step is to avoid the algorithm getting stuck at local optima. If the mutation will result in the creation of an individual with a low chance of survival, then the offspring produced will be discarded, but if the mutation creates a highly fit individual, it will improve the genetic characteristics of the population. Usually, mutation operator affects only a few genes (even only one gene as seen in Equation (3)) of an individual.(3)$$mutated\_individual=\left\{\begin{array}{c} gene\left[0\right],\;gene\left[1\right],\ldots gene\left[k-1\right],\;random\_number,\\ gene\left[k+1\right],\;\ldots gene\left[n-1\right] \end{array}\right\},$$
where *gene* [0…*n* − 1] represent the components of the individual to be mutated, the index *k* (which is randomly chosen in range [0…*n* − 1]), represents the position of the gene to be changed, and the *random_number* represents the value (randomly chosen) used to updates the gene with index *k*, and the curly brackets represent the concatenation operator.

The algorithm stops iterating either when a certain number of iterations has been performed, meaning that a certain number of simulations have been created and evaluated, or when an individual with “good enough” genes has been found, meaning that a model close to the optimal solution has been found within a given epsilon.

In current approaches, each individual represents a sequence of input stimulus groups to be provided to the DUT inputs. Each group of input stimuli is equivalent to a gene of the individual. As will be see below, the DUT used as a case study in the current paper has six inputs of one bit width. Therefore, each gene contains a group of six bits, as can be seen in Figure 2.

Each individual provides all the information needed to run a simulation on the DUT. At each clock cycle, the information carried by each gene is sent to the inputs of the DUT. At the end of the simulation, the simulation report is read and the functional coverage value is extracted. In the current situation, the fitness calculation is related to the functional coverage value: the individuals that obtained the highest coverage value are the most valuable individuals (Figure 3). For this reason, only children with the highest coverage values are selected to be the parents of the subsequent generation (this decision represents the selection process of the genetic algorithms).

During each generation, parents are combined using the crossover operator, as shown in Figure 4. In this way, new simulation scenarios are created, which brings the DUT into new operating states and could increase the value of functional coverage.

In the current work, the mutation affects only one gene in an individual. Mutation is very important because changing the values of a group of input stimuli can change all the states that will be reached by the DUT in the following simulation clock cycles. Thus, completely new scenarios can be created. These new situations create opportunities for the genetic algorithms to break out of the local minimum performance and reach the global maximum by achieving the highest possible coverage value. The mutation process is shown in Figure 5.

#### 3.1.2. The Custom Algorithms Developed Taking into Account the Principles of Genetic Algorithms

In order to develop sequences of input stimuli that can achieve high coverage, general principles of genetic algorithms are applied. Based on the Simple Genetic Algorithm (SGA) described in [20] (which was also addressed in [21]), the steps shown in Figure 6 were implemented, and the initial implementation was subsequently adapted to binary number processing and was improved to be more suitable for solving the current problem. In the first instance, several random data sets (having the structure required to be driven to the DUT inputs) are generated. The number of individuals per generation is passed at the beginning of the genetic algorithm using the *n_pop* parameter (Figure 6). The data created is structured as individuals.

In the second step of the process, the performance of the generated stimulus groups is evaluated during simulations. This is completed by creating text files containing stimuli to be driven to the DUT input. These files are read by the sequences called by the verification environment and passed on to the driver of the active verification agent. In this way, the stimuli are serially transmitted through the DUT inputs and simulation takes place. During the simulation, the values of the DUT outputs are recorded, and at the end of the simulation the coverage score is calculated. In this way, the initial individuals containing input data are correlated with the coverage value and an evaluation is implicitly obtained: the individuals that obtain the highest coverage value have the greatest potential in generating other high-performing individuals.

Next, corresponding to step 3 in Figure 6, individuals are arranged in descending order of the calculated scores to facilitate the execution of the next steps of the algorithm. Only the first half of the individuals in the sorted array will be used to generate new individuals. In addition, the data belonging to the sequence with the highest performance is saved to disk (step 4). A list that will be filled with children resulted from current generation is also initialized. The best *r_performant* % parents (*r_performant* is one of the parameters of the developed algorithms, *r_performant* ≤ *n_pop/*2) are copied to the list of children to avoid the risk of losing the best results from one generation to the next one. Based on the obtained score, individuals are selected and combined using the genetic crossover operator, with the aim of generating new better individuals from one iteration of the algorithm to the next (step 6). In addition, some of the resulting offspring are mutated (mutation is another genetic operator) to produce a slightly different situation from the normal evolution of individuals (step 7). The resulting individuals are added to the list of children of the current generation (step 8), and at the end of the algorithm iteration, the children are transferred to the list of parents to be used in the next iteration of the algorithm (step 9). The above steps are repeated for the number of iterations requested by the verification engineer. At the end of running the whole algorithm run, the parameters of interest of the evolution process are displayed (step 10). Several versions of this algorithm, depicted in Figure 6, were tested during the development of the work, as can be seen in this paper.

The randomness (existing in the initial data generation, in the choice of the crossover point and in the application of the mutation mechanism) combined with the application of genetic algorithms can provide good results in the automatic generation of high-performance sequences of input stimuli, thus reducing the time required for the functional verification of a module.

The crossover process is performed between parent 1 and parent 2, parent 2 and parent 3, parent 3 and parent 4, and so on. The children resulting from the crossover process are affected by mutation if Equation (4) is true, where rand () function gives a real number between 0 and 1. The higher the generation index, the higher the probability of mutation.
rand () < iteration_index/no_of_iterations,(4)

Several versions of genetic algorithms have been developed. The differences between them consist of the values of the configuration parameters and in the way the parents are processed to obtain the children. Each approach is tested on 40 generations of individuals, with each generation keeping the best 20 individuals from the previous iteration of the algorithm. Numbering the developed approaches does not provide additional information to the reader, but it has helped the authors keep track of differences between the algorithms used. The differences between the versions developed are mentioned in Table 1:

In V3.2 and V3.21 approaches offspring identical with their parents are ignored. If, at the end of any iteration, the number of generated individuals is not equal to the number of individuals of the initial population, other randomly generated individuals are added to the population. In this way, the population is rejuvenated, receiving new possibilities to get out from local maximums of performance.

In V3.3 and V3.6 approaches it is more likely that multiple copies of the same best element will occur. In this case, the cross-over process can often generate offspring identical with the parents.

#### 3.1.3. Established Genetic Algorithms Used for Comparison Purposes

In order to evaluate the performance of the approaches developed in the current study, the main coverage objective was also addressed using two of the most popular and widely used evolutionary algorithms [22]: the Nondominated Sorting Genetic Algorithm II (NSGA-II) and the Strength Pareto Evolutionary Algorithm 2 (SPEA2). The contribution of these algorithms can be seen in the automation of many computational processes, which is the goal of many research efforts worldwide [23,24].

NGSA-II is a computationally efficient multi-objective evolutionary algorithm (MOEA) based on a non-dominated sorting approach [17]. NGSA-II has a wide range of practical applications, being used in optimization of machining process parameters [25], in efficient organization of activities given the Generation Expansion Planning (GEP) problem [26], in medical diagnostics [27], in minimization of maximum completion time in the manufacturing industry [28], etc. In the present case, the algorithm was set to meet a single objective: increasing the coverage value of simulation runs obtained by generating appropriate stimuli.

NGSA-II uses a specific scheme for choosing pairs of individuals to be combined, which is based on non-dominated sorting and cluster distance comparison. The mechanism that implements this scheme is called tournament selection. The offspring resulting from the application of two of the genetic operators (outcrossing and mutation) on the parent pairs form the population of the next generation. Next, the resulting individuals are re-sorted using the tournament selection method and the selected ones become the parents of the next generation. In this way, most of the non-performing individuals are “lost” from one generation to the next, and the offspring generated at each iteration of the algorithm approach the set goal until it is reached.

In the current work, three approaches based on the NSGA-II algorithm were used (however, all implementations of the NSGA-II algorithm were adapted to search for a single-objective target instead of multi-objective targets, as needed by the study in this paper):A classical NGSA-II implementation, available in [29], in which the cross-over process consists of splitting the parent information using a single cross-over point; the children contain one section of data from the first parent and the other section of data from the second parent; the mutation operator modifies up to 5 randomly chosen bits in a data individual, according to Equation (5) (the result of this equation, which is 6 bits wide, is then rounded to the nearest integer; the *mutation_stength* is a parameter of the evolution process; the function *random* returns a real number between 0 and 1);Compared to the classical implementation, the mutation operator is changed to a version of Polynomial Mutation [30], as implemented in [31]; this version of the algorithm is hereafter referred to as “classical-modified NGSA-II”;Compared to the classical implementation, the mutation operator has been changed to Polynomial Mutation and cross-over operator was changed to Simulated Binary Crossover [32], according to the implementations presented in [31]; this version of the algorithm is hereafter referred to as “modified NGSA-II”.
(5)new_value=old_value−mutation_strength/2+random(0..1) × “mutation_strength”

Similar to NSGA-II, the second version of the Strength Pareto Evolutionary Algorithm (SPEA2) is one of the most powerful multi-objective evolutionary algorithms. “SPEA uses a mixture of established and new techniques to find multiple Pareto-optimal solutions in parallel” [33]. It was adapted to be used for single-objective objectives; therefore, the algorithm fits the requirements of the coverage objectives proposed in this paper.

SPEA2 is an evolutionary algorithm that has been successfully used in many domains. It helps to make a good decision in making investments by considering multiple objectives (profit, risk, etc.) [34], contributes to efficient management of resources (e.g., water) in distribution systems [35], improves communication systems in terms of average latency and average power consumption [36], reduces the amount of resources used in Field Programmable Gate Array (FPGA) devices [37] in the logic synthesis stage [38], and the examples can go on.

As for the fitness allocation scheme in SPEA2, each individual is assigned a fitness value computed based on the strength value of the dominating solutions. To distinguish between individuals with identical fitness values, additional density information is incorporated [18]. Each subsequent generation contains both the best parents of the previous generation and their children. The best individuals are characterized by the fitness value and the difference/set value compared to the other solutions [39].

In this study, the implementation of SPEA-II is based on the work of Professor Valdecy Pereira [40]. Unlike the goal in [40] where individuals are represented as real numbers, the functional coverage is given by a group of inputs driven to DUT ports. Therefore, the performance of a sequence of data elements must be evaluated and, consequently, the crossover and mutation operators used in [40] are not suitable for analysis in this paper. For this reason, the crossover and mutation operators used in the custom approach were also used in the approaches based on the SPEA2 evolutionary algorithm. During mutation, one of the individual values, which is randomly chosen, is changed to another value in the legal range.

### 3.2. Running UVM in Modelsim

The verification environment used in the current study follows the principles of the Universal Verification Methodology (UVM) [41] and is written in the SystemVerilog language [42]. The following commands need to be run to simulate the behavior of digital projects. The tool used for this task in the current work is the ModelSim^®^ simulator. Although the ModelSim^®^ simulator, developed by the Mentor Graphics company (now acquired by Siemens and integrated into the Siemens EDA division), does not support advanced verification capabilities (collection of coverage using coverage bins, execution of assertions and randomization of objects), it can still be used to perform verification according to the UVM methodology. To run a testbench containing components that inherit classes from the UVM 1.2 library, the following steps must be followed:After the ModelSim simulator is started, the directory must be set to the location of the Verilog/SystemVerilog sources;The library *work* is created;The following commands are issued: File -> Import -> Library-> Browse -> (go to ../ModelSim_ase/verilog_src/ and select the folder uvm-1.2)->ok->Next->Next->(Choose the destination folder containing the sources) ->Next->Next;The +define+UVM_NO_DPI flag is added to each compilation directive, as can be seen in following commands used to compile the sources;#compile uvm libraryvlog +define+UVM_NO_DPI uvm-1.2/src/uvm_pkg.sv# compile the designvlog +define+UVM_NO_DPI-reportprogress 300-work work design.svvlog +define+UVM_NO_DPI-reportprogress 300-work work actuator_interface.svvlog +define+UVM_NO_DPI-reportprogress 300-work work button_interface.svvlog +define+UVM_NO_DPI-reportprogress 300-work work sensor_interface.svvlog +define+UVM_NO_DPI-reportprogress 300 +define+WIDTH=8 +define+MAX_LIGHT_VALUE=900 +define+NO_OF_INTERVALS=20 -work work testbench.svThe *uvm_macros.svh* file must be included in the testbench (the top file of the project) and in the base test that is inherited in all verification tests;In many files in the imported uvm-1.2 library, paths to other files may need to be changed. An example of change is: from ‘include “uvm_hdl.svh” to ‘include “dpi/uvm_hdl.svh”.

Following the above steps, a verification environment incorporating legacy components inherited from the UVM library could be run in ModelSim^®^, given the limitations mentioned at the top of this section.

### 3.3. Using Genetic Algorithms-Based Approaches in Functional Verification

In order to use genetic algorithms to automate the fulfilment of functional coverage, data processed using genetic operators must be used during the simulation run. For this reason, a software-based processing system has been created, which is shown in Figure 7.

The general structure of the information processing ecosystem used in this paper and observed in Figure 7 includes the use of three languages: Python, Verilog, and SystemVerilog. The blue blocks refer to Python programming. The program starts with generating an initial population of input stimuli. Each group of stimuli is evaluated by being sent to the simulation environment in a regular structure (as text files). The simulation environment incorporates the verification environment (written in the SystemVerilog language), the DUT (coded in the Verilog language) and other structures needed to perform the simulations.

Given Figure 7, the designer(s) is/are responsible for building the DUT based on the functional specifications. They also create a simple testbench to verify the basic functionalities of the circuit. Verification engineers (whose number is usually larger compared to the number of designers) are in charge of creating the verification environment and a testbench (referred to in Figure 7 as “simulation environment”) used to link the DUT to the verification environment. The verification environment contains agents [43] that are tasked with feeding stimuli to the DUT via its input ports (active agents are implemented) and reading the design outputs (passive agents are implemented). One of the representative components of the active agents is the stimulus driver (visible in Figure 7), and one of the representative components of the passive agents is the monitor that helps collect coverage. In addition to agents, the verification environment contains reset handlers, sequence libraries, scoreboards and other components. However, these elements have not been represented in Figure 7 in order to keep the picture clean and representative for the currently de-written controller.

The process controller initiates a simulation each time the information in the input text files is replaced with new stimuli by the data generator whose operation is based on genetic algorithms. After each simulation, the value of the functional coverage is correlated with the set of stimuli that generated it. If the coverage value is higher, the stimulus set that generated it is considered more efficient. Genetic algorithms select the best individuals from each generation to develop them further using the crossover and mutation operators. A similar system that provides the ability to start hardware simulations from a Python script can be reviewed in [44].

The effectiveness of the method proposed in this paper and the differences between different versions of its application can be seen by analyzing the case study consisting of a circuit used for railway management inside a zoo.

### 3.4. Automation of the Verification Process

Genetic algorithm-based approaches, which are developed during current work, are used to generate stimuli to be sent to the input ports of the DUT. The programming system is built using the Python language. Additionally, by issuing commands to control the Modelsim^®^ simulator [45], the Python code is responsible for starting the design simulations and creating the command file (the command file has the extension.do and is generated by a simulation control panel) that is used to set up the simulations (files to be compiled are selected and test bench parameters are set). In addition, software written in Python is responsible for processing the simulation reports, extracting coverage values and correlating the stimulus sets with their corresponding scores. The entire information processing flow developed in the current study is shown in Figure 8.

The process of obtaining stimulus sequences that can achieve a high coverage value is initiated by a Python script after several important parameters of the genetic algorithms (number of iterations to be performed, number of individuals per population, and number of best individuals that are copied directly to the next population) and of the DUT functionality (number of sensor values contained by each individual and width of the sensor values) are determined. Next, iterations of the genetic algorithms follow each other until the number of iterations to be performed is reached. During the iterations, the score of each individual in the current population at that time is evaluated by running a simulation using the respective stimuli. The stimuli are read by two sequences (a sequence for data related to sensor values and a sequence for data related to button presses) in the verification environment from text files output by the Python program. The parameters needed for the simulation (e.g., width of sensor values, number of ranges, etc.) are written to a command file (randomly named run.do) that will be called by a batch script initiated by the Python program containing the *vsim -c -do “run.do”* command.

After compiling the files for the verification environment and the DUT, the simulation is run. When finished, a simulation report is created. Using regular expressions in Python, coverage information is extracted from the report and used as the fitness value output for the individual whose content was used to generate the stimuli in the current simulation. After the completion of each generation, the text files containing the best performing stimuli are saved to disk. The files resulting from the last iteration of the algorithm always contain the best performing stimulus sequences. These files can then be used either in stand-alone simulations or integrated into regressions used for detailed verification of a digital design.

### 3.5. DUT Description and Coverage Target

The DUT consists of an electronic management system for train entry within a railway section of a Zoo Garden. There are six trains with different routes. All of these share a section of track (hereinafter referred to as High Interest Railway Section—HIRS), as shown in Figure 9.

All trains may apply for entry in the high-interest section at each simulation clock cycle, but access is granted according to several rules:Traffic is controlled by two semaphores: if an odd-numbered train entered the HIRS, the semaphore for T1, T2, and T3 is green, and the semaphore for T2, T4, and T6 is red. If an even-numbered train arrives on the line, the reverse is true. If the section of track is empty, both traffic lights are green.If a train of one parity entered the HIRS, and in the next time slot priority is given to a train of a different parity, the rail section must remain empty for a time slot. Afterwards, the priority scheme is evaluated again.Odd numbered trains have priority over even numbered trains.Lower numbered trains have priority over higher numbered trains of the same parity.

Each HIRS state is characterized by the train that has gained access to enter it. The possible succession of HIRS states can be seen in Figure 10.

The DUT has an input pin associated with the request signal of each train. The main coverage objective (a simpler coverage objective is also shown in a section below) is to record that each train (there are six trains) has passed through the high interest section at least three times before 25 clock cycle arbitration runs. In addition, the HIRS must be empty for at least three clock cycles. Therefore, at least 21 HIRS access management clock cycles are required to achieve the maximum coverage value. If a sequence contains 25 transactions, it means that 4 transactions are kept as a buffer for cases where the DUT does not reach a state that contributes to meeting coverage. The positive edge of the arbitration clock cycle (marked as clock in Figure 11) represents the time when train requests are evaluated to calculate which one will be allowed to access HIRS.

Each group of input stimuli used in the data sequences incorporates six bits, one for each train. A value of one means that the train in question is requesting to enter the railway section of high interest and a value of zero means that the train is not interested in accessing the railway section in question. The *current_state* signal records at each point in time which train has gained access to the HIRS. Figure 11 shows a sequence of input stimuli sent to the digital railway management design.

The *read_inputs* signal is a DUT input automatically generated by the verification environment. the *even_semaphore* and *odd_semaphore* signals indicate which train categories (odd-numbered or even-numbered) are prohibited from accessing the track section of interest in the next clock cycle.

## 4. Results

In order to compare the performance of both the currently developed approaches and the classical method of performing functional verification, numerous simulations were performed using the ModelSim^®^ simulator. All simulations were performed using a system based on an Intel^®^ Core™ i7-3770S processor with a 3.10 GHz base frequency. The 16 GB of (Random Access Memory) RAM available was sufficient to satisfy the need to run multiple genetic algorithm-based approaches in parallel.

### 4.1. Analysis of the Results in Terms of the Main Coverage Target

The performance obtained by each approach, when the resulting children from the crossover process are moved according to (1), can be seen in Table 2. The results obtained using the developed approaches, which are the subject of this paper, are compared with the results of the NSGA-II and SPEA2 algorithms. The SPEA2 algorithm was run twice over 40 iterations. Thus, it can be seen that the randomness involved in generating the initial population, performing the mutation and choosing the crossover points can strongly influence the performance of the training process. In order to justify the necessity of using genetic algorithms, 800 simulations were also performed using random stimuli. These simulations are the exponent for the classical way of stimulus generation used in the functional verification process (that is known as constrained-random generation). In case of random generation, only one set of stimuli reached the maximum coverage value. Therefore, since the coverage target is not easily reached, the need to implement automation mechanisms becomes obvious.

Near the section “the number of stimulus sets that achieve maximum coverage”, the number of root datasets is mentioned in parentheses. In the case of custom algorithms, many more stimulus sequences are generated, but all are very similar, in most cases only a few elements are different. Therefore, the root datasets represent the “pattern of values” that is inherited by multiple stimulus sequences.

Given the results in Table 2, it can be understood that the performance of genetic algorithms is also influenced by the randomly generated initial population. The fact that the V3.3 approach, the classical modified NSGA-II algorithm and the SPEA2 algorithm (during the second trial) reached 100% coverage of achievement as early as the second or third iteration is closely related to the amount of randomly generated stimuli and is not a result of the evolution process. However, due to the evolution process performed during subsequent iterations, all tested approaches managed to reach the maximum coverage value no later than the fifteenth iteration.

An intuitive way to evaluate the performance of the developed versions of the algorithm is to plot both the coverage achievement and the number of stimulus sets that reached the maximum coverage value at each training iteration. In all images, the number of stimulus sets (orange) represents the number of stimulus sequences that have the maximum score found no later than the iteration whose index is marked on the x-axis.

In the case of Currently Proposed Approaches (CPA), starting with iteration 5, all approaches provided at least one sequence that resulted in reaching the maximum coverage value. Figure 12 shows a balanced evolution of sequences over the algorithm iterations. The maximum coverage value gradually increases as more and more target coverage intervals are reached. In contrast to these results, apart from the classical modified NSGA-II case, in which high-performing stimuli were well generated even from the initial population, NSGA-II algorithms fail to construct the first sequence of high-performing stimuli only in the ninth iteration. The SPEA2 algorithm is found to perform similarly to CPA, considering the first obtained high-performing stimulus sequence and the first-generation index containing 10 high-performing sequences. However, the number of performing data sequences generated by the SPEA2-based approach is larger than the number of sequences generated by the approaches developed during the present work, which are based on the implementation of the simple genetic algorithm in [20]. The main observation is that the CPA implementation is simpler and easier to understand than the NSGA-II and SPEA2 algorithms.

Considering Figure 12, some interesting observations can be drawn. Genetic algorithms encourage a process of evolution of individuals. In the current situation, evolution consists of generating sets of stimuli that reach a higher coverage value, derived from individuals with a lower coverage value. Figure 12d shows a good example of gradual evolution. After the first set of stimuli reaching the maximum coverage value has been reached, the approaches used continue to generate new sets of stimuli. In the case of CPA, most of these are similar to the first set of high-performing stimuli generated. Figure 12c shows well how the number of HPSS increases as training progresses. The target number of HPSS is a good candidate for the stopping condition of a genetic algorithm-based approach. Considering only the first ten training iterations depicted in Figure 12, the proposed approach V3.3 generated nine HPSS, the SPEA2 algorithm generated eight HPSS (Figure 12h), V3.2 generated six HPSS (Figure 12a), and all other approaches generated less than six HPSS (Figure 12b,d–g).

Considering only custom developed approaches and trials using the SPEA2 algorithm (NSGA-II is excluded from this analysis as it does not use a mutation probability parameter), if the mutation coefficient is set to 0.05 (an appropriate value considering [46]) for all generations, the results in Table 3 are obtained.

Additionally, a visual description of the algorithm performance can provide easy-to-understand information about the learning capabilities of each tested approach, as can be seen in Figure 13.

The V3.2 approach was found to be of low quality in this context, as although it achieved 100% coverage from the third iteration onwards, it was not able to generate more than two HPSS until the tenth iteration (picture a in Figure 13). In the case of the V3.21 approach (Figure 13b), four new HPSS were generated in the tenth iteration. This observation exemplifies the “parallel” behavior of genetic algorithms: several pairs of individuals are combined at the same time using the crossover operator, and each of these pairs can independently generate new HPSS. The V3.6 approach was misconfigured and was not able to learn at all (Figure 13d). Usually, the reason for such a situation is the misconfiguration of the communication environment between the algorithms used for automation and the employed simulator. In Figure 13e,f, a continuous increase in the number of HPSS can be observed. This is a characteristic of a correct training process.

The performance of approaches V3.3 and V3.6 seems to be low for the fixed value of the mutation coefficient. However, randomly generated populations also influence the future performance of each genetic algorithm. Since half of the CPAs did not reach 100% coverage, it can be concluded that it is more convenient to increase the mutation coefficient with the generation index, rather than using a fixed value. This conclusion is confirmed by the column in Table 2 and Table 3 showing the number of stimulus sets reaching maximum coverage: a higher number of sequences reaching maximum coverage was obtained when the mutation coefficient was changed during the training process than when the mutation coefficient was linked to a fixed value.

For SPEA2 algorithms, proportionally updating the mutation coefficient according to the generation index reduced the number of high-performing sequences during the training process, which consists of running 40 iterations of the algorithm. This observation was expected, since a higher mutation coefficient (which is obtained when updated from 0 to 1) leads to finding more similar results. In addition to this, both approaches reached the first high-performing stimulus sequence (HPSS) at the latest at the sixth iteration. In contrast, when the mutation coefficient was tied to 5%, the first ten HPSS were obtained faster compared to when the mutation coefficient was updated proportionally to the index of each generation.

Figure 13 shows the first 10 training iterations when the mutation coefficient is set to 0.05. Figure 14 expands on the information presented in Figure 13, providing an overview of the entire training process.

All training processes depicted in Figure 13 gradually increase the number of HPSS during training iterations. The behavior of the genetic algorithms is not constant during training iterations. As can be seen in the case of CPA (Figure 14a–d), there are generations inside where the number of stimulus sets remains at the same value and generations in the same training processes when the number of HPSS increases even by more than one unit. Additionally, the effect of randomness that characterizes genetic algorithms is well represented in Figure 14e,f. The same algorithm, when run a second time, doubled its performance.

### 4.2. Analysis of the Results in Terms of an Easier Coverage Target

An achievable coverage goal related to the DUT functionality in question is to require each train to gain access to HIRS at least once per simulation when running a sequence of seven input combinations in each simulation. For this request, a sequence of 200 randomly generated stimulus groups was generated and run. This sequence is the exponent of the classical way of applying functional verification. After running the sequence, the 100% coverage value was met. Thus, for this simple coverage goal, if only one HPSS is needed, there is no need to use an advanced verification automation mechanism (e.g., genetic algorithm-based methods). However, for exemplification, genetic algorithm-based approaches were tested. In the case of CPA, the mutation coefficient was calculated according to (1) and a predefined number of best parents are copied to the next generation. The results for the versions of genetic algorithms listed can be seen and compared in Figure 15.

For the second coverage target, an interesting situation arises. Several models seem to be “resistant” to the generation of new HPSS. In Figure 15a,b,e,f,h, one can observe large periods where the number of stimulus sets remains constant. In contrast, approach V3.6 (picture d) has an almost linear increase in number of HPSS. This attribute is one of the factors that makes this approach a winner in solving the second coverage objective. For this target, approaches based on the NSGA-II algorithm have a low average performance both in terms of the number of stimulus sets that reach maximum coverage and in terms of the number of iteration in which the maximum coverage value is finally reached. Approach V3.3 (Figure 15c) and approach V3.6 (Figure 15d) have only one difference: in the first mentioned approach, the best performing parents are copied twice in the next generation and in V3.6 approach the parents are copied only once. This aspect, combined with the randomness of the genetic algorithms, make the V3.3 approach the least performing and the V3.6 approach the best performing. For this reason, when developing a new genetic algorithm-based approach, it is worth configuring it in several ways to find the setup that best fits the functionality of the targeted digital design.

There are also important parameters that have not been explicitly mentioned in the figures to keep them clean and light, but whose values have been recorded in Table 4.

Considering Figure 15, apart from V3.3, all versions of the genetic algorithm-based approaches achieved 100% coverage of satisfiability, thus reaching the coverage target. However, the graph showing the performance of the V3.3 approach is a good example of the benefits of using genetic algorithms: even if the maximum coverage value is not reached, the number of high performing stimulus sets increases during the training process. In view of Table 4, one aspect related to the number of stimulus sets generated should be commented. By far, the CPA and SPEA2 algorithms can generate many similar data sequences, while NSGA-II based approaches generate fewer but more diverse data sequences. The V3.6 approach was able to obtain the highest number of high-performance stimulus sets, and the modified NSGA-II algorithm generated the highest diversity of stimulus sets.

According to Table 4, the V3.21 and V3.6 approaches reached the coverage goal significantly faster (in the third and fifth iteration) than the NSGA-II algorithm-based approaches (in the eleventh, twelfth, and seventeenth iteration) or the SPEA2 algorithm-based approaches (in the eighth iteration).

In terms of the current coverage target, in order to find only a solution with 100% coverage, it is not necessary to use genetic algorithms. The 100% coverage was also achieved by randomly generating 200 stimulus sequences. Therefore, this coverage goal can easily be achieved using only the common flow of functional verification. Genetic algorithm-based approaches become useful in this case if the target is to generate many stimulus set achieving 100% coverage. The study in this Section 4.2 represented a simpler case of coverage goal. In other cases, purely random simulations can consume a large amount of time before the coverage goal is reached.

## 5. Discussion

In this paper, a software system used to automatically obtain stimulus values from input files, to start simulations, and to read the simulation results was presented. The development of this system facilitated the application of genetic algorithms in reducing the time of the coverage process. Verification engineers are only tasked with setting up the algorithm, determining the number of iterations, choosing how to apply the crossover process, and selecting the next generation members. With little effort, this approach can be used for other methods of automating the functional verification process.

The first step to evaluate whether it is worth using genetic algorithms to create an advanced input stimulus generation mechanism is to test the result obtained by randomly generating input stimuli. If the randomly generated stimuli fail to achieve an adequate level of coverage in a timely manner, it is necessary to use more advanced generation mechanisms. The second step is to estimate the time and effort required to adapt a genetic algorithm approach to the current verification process. Finally, the costs of using the simulation infrastructure must be evaluated in both cases. After this analysis, the verification team can make an informed decision on the verification approach to be used. In this paper, for each coverage objective, the appropriateness of using genetic algorithms has been analyzed. In the case of the main coverage target, 800 simulations incorporating random stimulus generation were performed. Only one simulation achieved 100% coverage. For the easier coverage target, by running 200 simulations incorporating random stimulus the 100% coverage target was reached once and by running 800 simulations three sets meeting 100% coverage fulfillment were generated. That shows that, for the second coverage target, it is not worth investing time in establishing more sophisticated automation methods for generating only one high-performing stimulus set. However, genetic algorithm-based approaches are appropriate if multiple sets of inputs have to be generated.

As can be seen in Table 2, all custom-developed genetic algorithm-based approaches achieved 100% coverage at the latest by the fifth iteration. The V3.6 approach, after performing at least 800 simulations (at least 20 simulations/iteration × 40 iterations), succeeded to generate 205 different stimulus sets achieving 100% coverage, while, during 800 simulations, the random constrained generation method supplied only one high performance group of inputs. NSGA-II-based approaches were slower in generating a stimulus set that achieved the maximum coverage value: with the exception of the classical modified NSGA-II, where the initial population already contained a high-performing individual with a score of 95.2% (that in the second iteration produced a stimulus set obtaining 100% coverage), the other algorithms did not reach the coverage achievement target until the ninth or fifteenth iteration. Therefore, the approaches developed in the present work were found to perform better than NSGA-II in terms of speed of constructing a high-performing dataset. The SPEA2 algorithm-based approaches performed better than the NSGA-II-based approaches and performed similarly compared to CPA, given the iteration index in which the first HPSS and the first 10 HPSS were generated). In contrast, using the classical verification method (represented by randomly generated input stimuli), only one high-performing stimulus set was found in a group of 800 randomly generated data sets. Therefore, the classical stimulus generation method was outperformed by all genetic algorithm approaches

The performance of genetic algorithm approaches is also highlighted by the number of stimulus sets generated that reached maximum coverage. On the one hand, according to Table 2, the NSGA-II and SPEA2 algorithms were able to generate a higher diversity of datasets than CPA. On the other hand, the custom developed approaches obtained a very large number of similar datasets (CPA obtained more datasets than NSGA-II based approaches, but fewer datasets compared to SPEA2 approaches). Although these stimuli were obtained by mutation mechanisms and not by exploiting the true benefits of the crossover operator (which in other situations may provide groups of stimuli with less similarity compared to the mutation operator), the obtained datasets can still be used to detect errors in the verified design.

Another interesting observation was obtained by analyzing Table 2 and Figure 12. The V3.3 approach, the modified classical NSGA-II algorithm and the SPEA2 algorithm (during the second attempt) achieved 100% coverage as early as the second or third iteration, because the randomly generated stimuli were very well distributed over the coverage intervals. Therefore, the use of genetic algorithms continues to benefit from the advantages of random simulations, while ensuring that the evolution of stimulus sets will increase their performance.

Genetic algorithms can provide good results if their parameters are set appropriately. There are two ways of understanding what the appropriate parameter values should be for a specific use case of genetic algorithm-based approaches. The first is to search the literature for values commonly used in practice. In this way, a solid background is created by consulting the experience of other verification engineers. Second, once several suitable values for the configuration parameters are found, they must be tested on the specific verification case to be addressed. In this way, the most efficient algorithm will be selected to increase the performance of the functional verification process from several well-configured approaches. This paper has addressed these two steps, firstly by consulting other related industry initiatives and using the experience of other verification engineers (e.g., using the 5% value for the mutation rate threshold) and secondly by configuring the infrastructure created using genetic algorithms (Figure 6) in several ways (Table 2, Table 3 and Table 4).

Although the approaches currently used are different, in some cases they have produced similar results. According to Table 2, one of the similarities between the performances of the mentioned approaches is that the first-generation index, when at least 10 stimulus sets reach the maximum coverage value, has values mostly between 11 and 15. Another interesting observation is that the V3.6 approach, in which the best performing parents are copied only once in the next generation, obtained the highest number of stimulus sets reaching maximum coverage considering CPA. In conclusion, V3.6 is considered the most suitable CPA to be used for the complex coverage goal presented in this paper. In contrast, according to the CPA data in Table 3, V3.21 was able to generate the highest number of high-performing datasets (having two different roots, which is a positive thing) when the mutation threshold was 5%. The properties of an approach can make it more useful for one situation and less useful for another. It is therefore necessary to develop and test multiple versions of genetic algorithms when testing a DUT. It should also be noted that when the mutation rate was bound to 5%, the SPEA2 algorithm outperformed the CPA, given the number of stimulus sets that reached the maximum coverage value and the first-generation incorporation index of 10 HPSSs.

Considering Table 2 and Table 3, the choice of mutation rate at 5% had a negative effect in V3.21, V3.3, and V3.6 and a slightly positive effect in V3.2. Considering the evolution of the V3.2 approach, the same level of coverage was reached in both mutation rate configurations, but when the 5% value was used, three different roots of stimulus sets were generated instead of one for the situation where the mutation threshold had a mobile value. Thus, although sometimes a fixed value for the mutation rate can provide good results, a dynamically updated value of the mutation rate during the training process is considered a provider of better performance. Additionally, although there are many complex algorithms that deal with dynamically changing the mutation rate for different applications of genetic algorithms [47,48,49], in this paper a simple rule for updating the mutation rate has been developed, as can be seen in Equation (4). Its performance has been proven in most of the approaches presented in this paper, which makes Equation (4) one of the welcome elements introduced by this paper.

Additionally, it can be seen in Table 2 and Table 3 that the average time for achieving the best results is shorter for custom developed approaches than for NSGA-II and SPEA2 approaches. However, the relevance of this observation is low from the point of view of evaluating the performance of the algorithms, since the number of simulations run by each genetic algorithm-based approach during each training iteration is similar (approximately 20 simulations are run per iteration, where 20 represents the initial population size). The time differences can be explained given the state of the computing machine: if it has to perform several tasks simultaneously on the same CPU, all tasks will last longer.

In the current work, the most computational time spent by each process is consumed by running ModelSim^®^ simulations. The time consumed by the rest of the automation system (which is written in Python) is considerably less. Therefore, we consider that the processing time should be similar for each algorithm addressed in the current study. However, an interesting observation from Table 3 is that V3.21 approach managed to reach the desired results in the shortest time, although it needed more iterations than SPEA2-based approaches. This observation suggests that while running SPEA2-based approaches, the CPU had to perform more tasks in parallel. Therefore, in addition to the efficiency of the automation approach, the state of computing machine greatly influences the time needed to achieve the desired results.

Related to the analysis of the approaches developed to achieve the second coverage achievement goal, the V3.6 approach outperformed all other training runs because it generated the first HPSS and the first 10 HPSS in the fifth and thirteenth iterations. The V3.6 approach also generated the highest number of HPSSs (172 stimulus sets) in 40 generations (each of them consisting in running of 20 simulations). The input stimulus groups, based on their elements, can be clustered in two groups of similar input stimulus sets. When using random generated stimuli, only three input sets achieving 100% coverage fulfillment were generated, after running 800 simulations. Therefore, the ratio between number of stimulus sets generated by highest performing version of genetic algorithm-based approach (for the second coverage target) and the number of stimulus sets generated by running random simulations is more than 57:1. Additionally, the modified NSGA-II algorithm was able to generate 12 totally different sets of sequences. This result can be sometimes more desirable because different sequences can bring the DUT into several interesting operating states that can reveal unexpected functional errors in addition to meeting the coverage goal. However, Table 4 supports the conclusion that CPA is a good alternative to the more popular algorithms in generation of stimuli, given its easier implementation.

## 6. Conclusions

One of the main contributions of the current work is the demonstration that a dynamic mutation rate value significantly improves the performance of genetic algorithms. It has been shown that Equation (4) represents a simple but efficient novel way to dynamically change the mutation rate value during algorithm iterations.

Automating the classical method of performing checks can save valuable work time. Automatically correlating stimulus sets with the appropriate age-coverage value and combining the best performing stimulus sets is shown to create an evolutionary mechanism that creates increasingly better individuals as generations succeed each other. Therefore, it has been shown that the classical way of performing verification can be improved by generating input stimuli using genetic algorithms. In this way, coverage closure is achieved in a shorter time compared to constrained random generation. Additionally, genetic algorithms are efficient in generating multiple derivations of high-performance input stimulus sequences that help to create a consistent test base needed for verifying digital designs.

Custom genetic algorithm-based approaches have been compared with two of the most popular evolutionary algorithms: NGSA-II and SPEA2. It has been shown that custom-developed approaches can provide similar results to those provided by these algorithms (when NGSA-II and SPEA2 algorithms are configured to have a single objective function). The stimuli obtained by all the presented algorithms can be further used to verify the correctness of digital design functionalities.

Genetic algorithms can give good results if their parameters are set appropriately. For each unique verification problem, there are several solutions available that need to be tested in order to choose the most suitable one.

The time taken to find a good solution should always be justified by the quality of the results obtained. Therefore, for each functional coverage fulfilment situation, an analysis should be performed to see whether random stimulus generation is sufficient or whether more advanced generation mechanisms should be applied. In the present study, the more complex coverage goal required automating stimulus generation using genetic algorithms. However, the second coverage goal did not warrant the application of genetic algorithms.

This paper presented an end-to-end communication system between a software-based controller and a hardware emulator (represented by the simulation environment). The program built using the Python language follows several easy-to-understand steps that provide an efficient mechanism for faster coverage fulfillment compared to classical verification based on constrained random generation. The designed system can be further developed and used in other approaches that focus on automating functional verification.

Given the results, this work joins the worldwide sustained efforts to achieve a shorter development time for electronic circuits by transferring part of the work of verification engineers to genetic algorithms, which are part of the pool of artificial intelligence automation methods. Additionally, this work used only free tools, with no loss in process performance. This is very important for start-ups, small research groups or students who intend to enter the world of functional verification based on modern automation methods.

## Figures and Tables

**Figure 1 micromachines-13-00691-f001:**
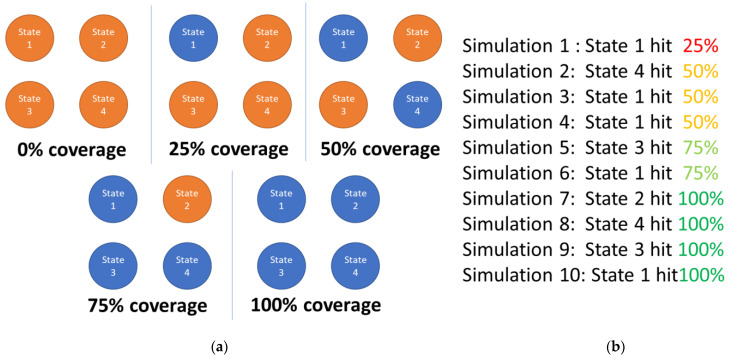
(**a**) Evolution of state coverage for an FSM; (**b**) list of states addressed by the FSM during each simulation.

**Figure 2 micromachines-13-00691-f002:**
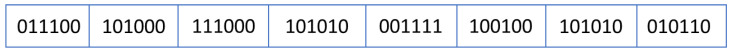
Representation of an individual with eight genes constructed according to the peculiarities of DUT in question.

**Figure 3 micromachines-13-00691-f003:**
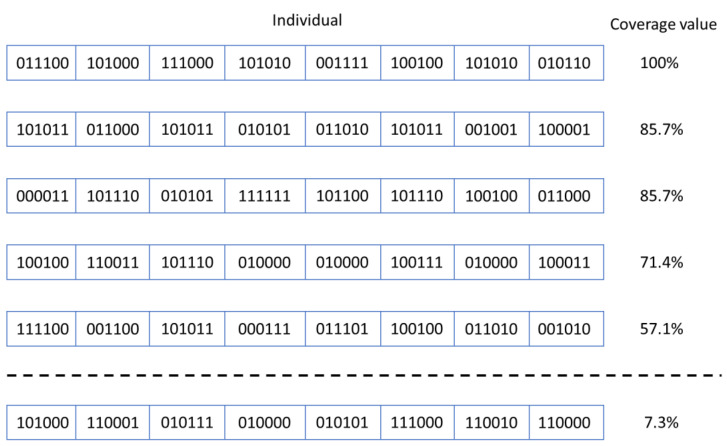
The individuals are ordered according their fitness function whose result is the coverage value.

**Figure 4 micromachines-13-00691-f004:**
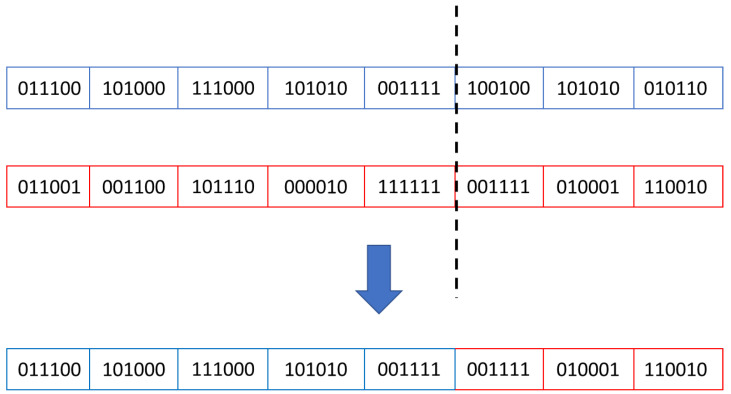
Representation of the crossover process. The crossover point (the point where individuals are split to exchange their second part) is randomly chosen during each crossover process.

**Figure 5 micromachines-13-00691-f005:**
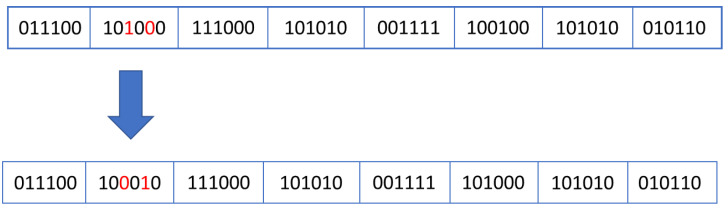
Representation of the mutation process. One element of the individual is randomly selected and its content is randomly modified.

**Figure 6 micromachines-13-00691-f006:**
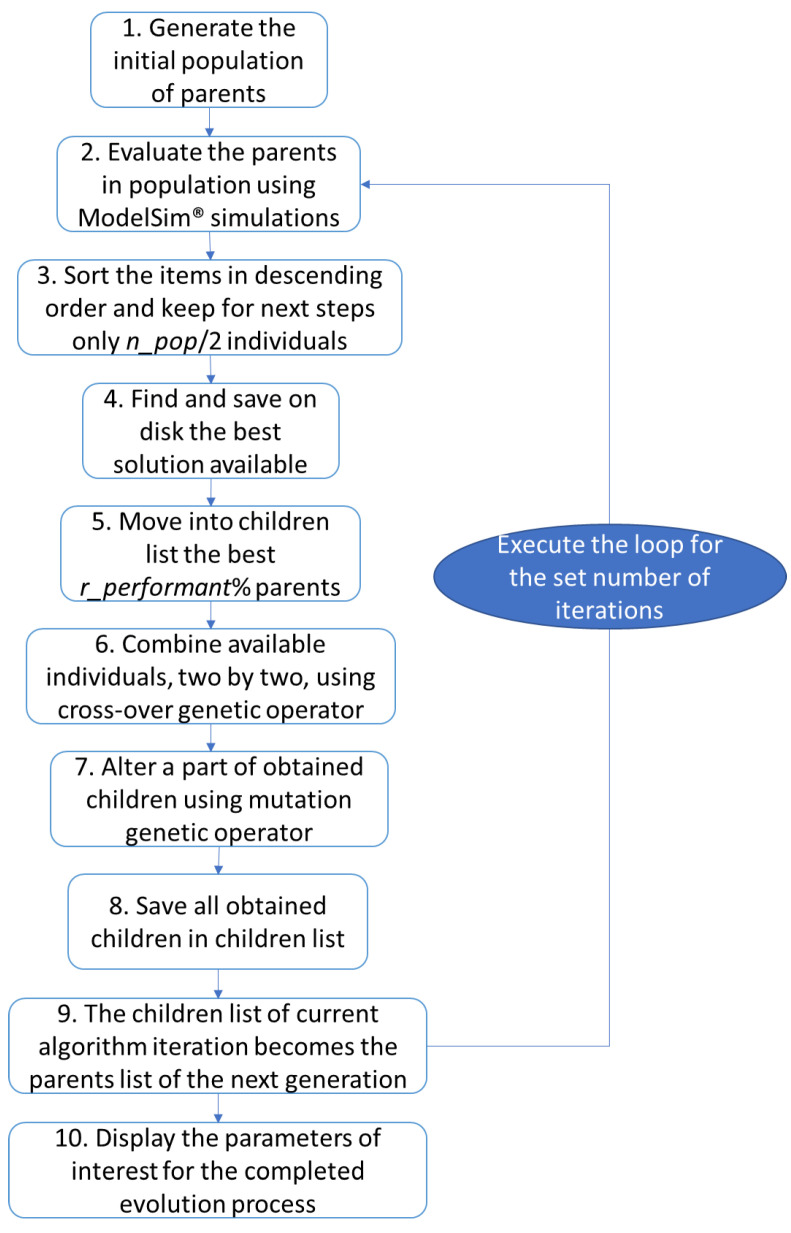
Flowchart of the general structure of the developed genetic algorithm.

**Figure 7 micromachines-13-00691-f007:**
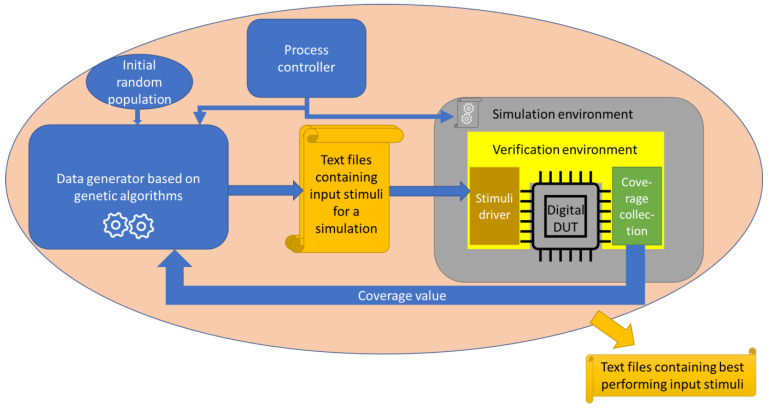
Schematic of the programming ecosystem used to generate high-performance input stimuli.

**Figure 8 micromachines-13-00691-f008:**
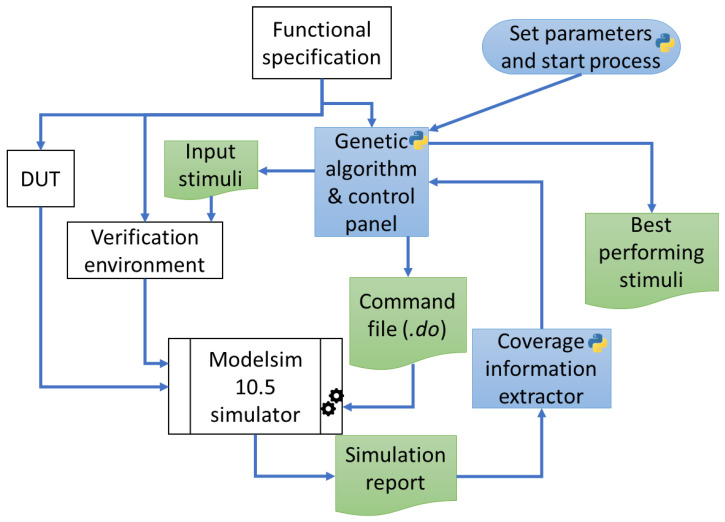
Information processing flow developed in the current research. The blue boxes represent modules written in Python, the green shapes represent text files generated during system operation, and the white boxes are items that are used during the classical verification process. Genetic algorithms tune the verification process by adding a layer of automatic control to the classical verification process.

**Figure 9 micromachines-13-00691-f009:**
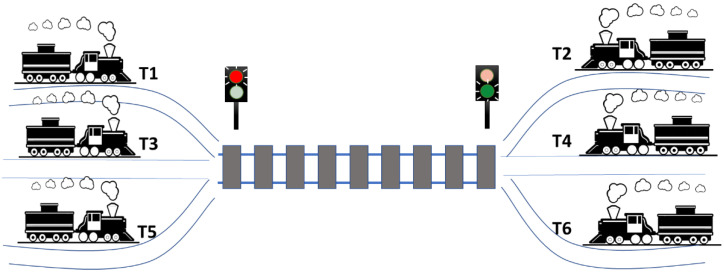
General idea of the DUT employed in the current work.

**Figure 10 micromachines-13-00691-f010:**
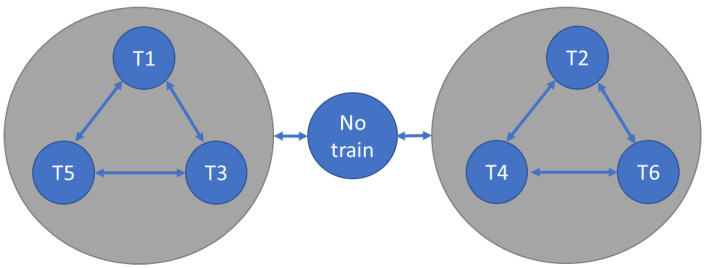
Possible transitions between HIRS states. Trains of different parity cannot pass one after another on the railway section of interest without leaving the section empty for a certain period (in this case, one clock cycle).

**Figure 11 micromachines-13-00691-f011:**
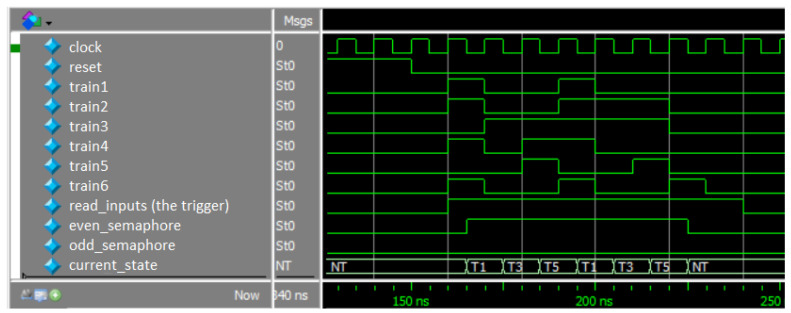
Screenshot from the Modelsim simulation of the digital railway management design.

**Figure 12 micromachines-13-00691-f012:**
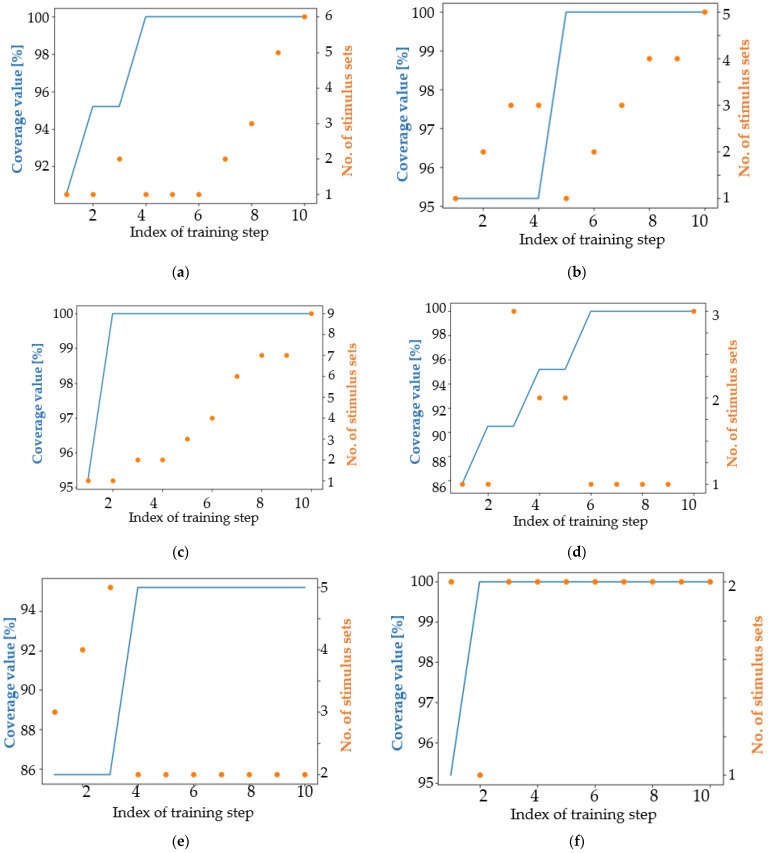
The first 10 iterations of training for each approach, with the primary coverage goal. For the custom developed approaches, the mutation coefficient was updated proportionally considering the number of iterations of the algorithm. The label “proposed” is attached to plots related to the approaches developed and proposed during current study. The blue line shows the coverage value reached by the best performing element in the population at that training step. The orange dots show how many individuals reached the highest coverage value met at each training step. For example, in sub-figure (**g**), at the ninth training step, the population contained only one individual reaching 100% coverage. For sub-figures (**a**,**g**,**h**), the best individuals in the initial population reached 85.7% coverage. For sub-figures (**d**,**e**), the best individuals in the initial population reached 90.5% coverage. For sub-figures (**b**,**c**,**f**), the best individuals in the initial population reached a coverage of 95.2%. (**a**) V3.2 (proposed); (**b**) V3.21 (proposed); (**c**) V3.3 (proposed); (**d**) V3.6 (proposed); (**e**) classical NSGA-II [29]; (**f**) classical-modified NSGA-II [31]; (**g**) modified NSGA-II [31]; (**h**) SPEA2 [40]—Trial I.

**Figure 13 micromachines-13-00691-f013:**
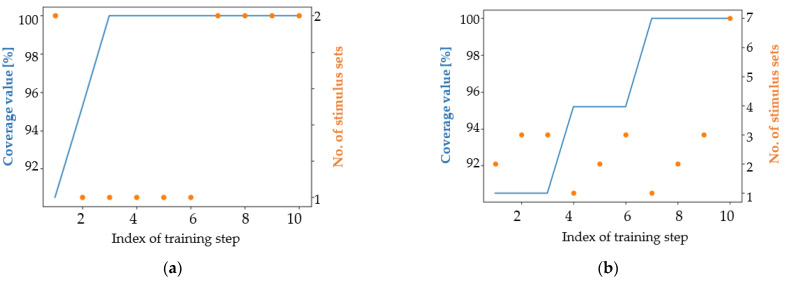
First 10 learning iterations for each approach considering the main coverage target when the mutation coefficient is 5%. The label “proposed” is attached to plots related to the approaches developed and proposed during current study. The blue line shows the coverage value reached by the best performing element in the population at that training step. The orange dots show how many individuals reached the highest coverage value met at each training step. For example, in sub-figure (**e**), at the seventh training step, the population contained three individuals reaching 100% coverage. For sub-figure (**c**), the best individuals in the initial population reached 81% coverage. For sub-figure (**d**), the best individuals in the populations at each training step reached 85.7% coverage. For sub-figures (**a**,**b**,**e**,**f**), the best individuals in the initial population reached a coverage of 90.5%. (**a**) V3.2 (proposed); (**b**) V3.21 (proposed); (**c**) V3.3 (proposed); (**d**) V3.6 (proposed); (**e**) SPEA2 [40]—Trial I; (**f**) SPEA2 [40]—Trial II.

**Figure 14 micromachines-13-00691-f014:**
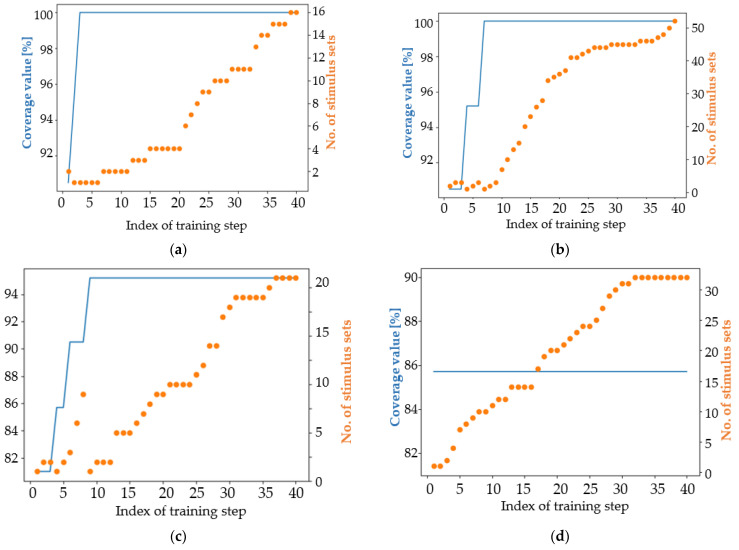
Training iterations when the mutation percentage equals 5%. The label “proposed” is attached to plots related to the approaches developed and proposed during current study. The blue line shows the coverage value reached by the best performing element in the population at that training step. The orange dots show how many individuals reached the highest coverage value met at each training step. For example, in sub-figure (**c**), at the sixth training step, the population contained three individuals reaching 90.5% coverage. For sub-figure (**c**), the best individuals in the initial population reached 81% coverage. For sub-figure (**d**), the best individuals in the populations at each training step reached 85.7% coverage. For sub-figures (**a**,**b**,**e**,**f**), the best individuals in the initial population reached a coverage of 90.5%. (**a**) V3.2 (proposed); (**b**) V3.21 (proposed); (**c**) V3.3 (proposed); (**d**) V3.6 (proposed); (**e**) SPEA2 [40]—Trial I; (**f**) SPEA2 [40]—Trial II.

**Figure 15 micromachines-13-00691-f015:**
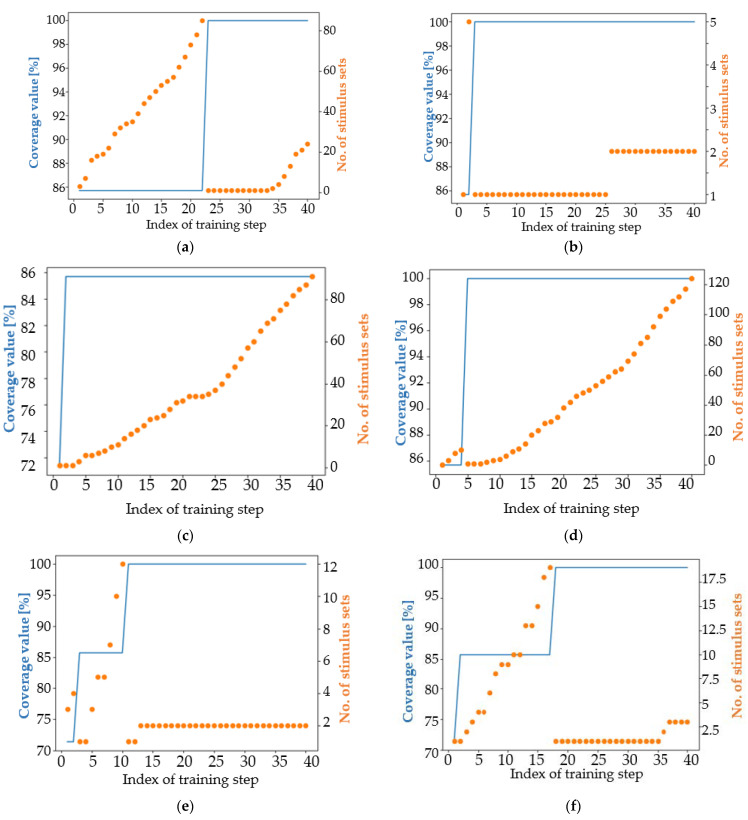
Plots representing the outcome of the genetic algorithm-based approaches used for the easy coverage target. The label “proposed” is attached to plots related to the approaches developed and proposed during current study. The blue line shows the coverage value reached by the best performing element in the population at that training step. The orange dots show how many individuals reached the highest coverage value met at each training step. For sub-figures (**c**,**e**,**f**), the best individuals in the initial population reached 71.4% coverage. For sub-figure (**a**,**b**,**d**,**g**,**h**), the best individuals in the populations at each training step reached 85.7% coverage. (**a**) V3.2 (proposed); (**b**) V3.21 (proposed); (**c**) V3.3 (proposed); (**d**) V3.6 (proposed); (**e**) classical NSGA-II [29]; (**f**) classical-modified NSGA-II [31]; (**g**) modified NSGA-II [31]; (**h**) SPEA2 [40]—Trial I.

**Table 1 micromachines-13-00691-t001:** Differences between versions of genetic algorithm implementations.

Algorithm Version	The Best Performing Parents Are Copied in the Next Generation	Offspring Identical to Their Parents Are Discarded
V3.2	twice	yes
V3.21	once	yes
V3.3	twice	no
V3.6	once	no

**Table 2 micromachines-13-00691-t002:** Tabular comparison between genetic algorithm-based approaches and a sequence of 800 randomly generated stimulus sets for the main coverage target.

Criterion	Proposed Approaches				Classical Approach	Established Genetic Algorithms				
	V3.2	V3.21	V3.3	V3.6	Random Stimuli	Classical NSGA-II (Adapted after [29])	Classical-Modified NSGA-II (Adapted after [31])	Modified NSGA-II (Adapted after [31])	SPEA2 ^2^ (Adapted after [40])	
									Trial I	Trial II
maximum coverage reached	100	100	100	100	100	100	100	100	100	100
parents/generation	20	20	20	20	800	20	20	20	20	20
the first iteration containing a dataset which leads to maximum coverage	4	5	2	5	1	15	2	9	6	3
time passed until first high-performing dataset is reached [minutes]	6 m	6 m	4 m	11 m	38 m	24 m	3 m	12 m	21 m	11 m
first generation when at least n_pop ^1^/2 elements having maximum coverage were obtained	12	12	11	13	n/a	27	n/a	n/a	11	15
time passed until n_pop/2 elements having maximum coverage were obtained [minutes]	19 m	17 m	24 m	26 m	n/a	44 m	n/a	n/a	38 m	53 m
the number of sets of stimuli that achieve maximum coverage	49 (1)	176 (1)	146 (1)	205 (1)	1	45 (1)	3 (3)	2 (2)	285 (2)	212 (3)

^1^ n_pop represents the initial number of individuals per population and is equal to 20 in current implementations. ^2^ in the case of the SPEA2 algorithm, the same algorithm was run twice and, due to the randomness involved in the process, different results were obtained.

**Table 3 micromachines-13-00691-t003:** Results related to the main coverage target when the mutation percentage was set to a value of 5% value for each iteration of the algorithm.

Criterion	Proposed Approaches				Established Alg.	
	V3.2	V3.21	V3.3	V3.6	SPEA2 ^2^ [40]	
					Trial I	Trial II
maximum coverage reached	100	100	95.2	85.7	100	100
parents/generation	20	20	20	20	20	20
the first iteration containing a dataset that leads to maximum coverage	3	7	9	1	6	4
time passed until first dataset with 100% coverage is reached [minutes]	6 m	10 m	n/a	n/a	22 m	12 m
first generation when at least n_pop ^1^/2 elements having maximum coverage were obtained	26	11	21	8	9	9
time passed until n_pop/2 elements having 100% coverage were obtained [minutes]	48 m	16 m	n/a	n/a	31 m	26 m
the number of stimulus sets that achieve maximum coverage	16 (3)	52 (2)	21 (1)	32 (1)	84 (1)	175 (1)

^1^ n_pop represents the initial number of individuals per population and is equal to 20 in current implementations. ^2^ in the case of the SPEA2 algorithm, the same approach was run twice and, due to the randomness involved in the process, different results were obtained.

**Table 4 micromachines-13-00691-t004:** Tabular comparison between genetic algorithm-based approaches run over 40 iterations and a sequence of 200 randomly generated stimulus groups.

Criterion	Proposed Approaches				Classical Approach		Established Genetic Algorithms			
	V3.2	V3.21	V3.3	V3.6	Random Stimuli		Classical NSGA-II (Adapted after [29])	Classical-Modified NSGA-II (Adapted after [31])	Modified NSGA-II (Adapted after [31])	SPEA2 (Adapted after [40])
maximum coverage reached	100	100	85.7	100	100	100	100	100	100	100
parents/generation	20	20	20	20	800	200	20	20	20	20
the first iteration containing a dataset that leads to maximum coverage	22	3	2	5	1	1	11	18	17	8
first generation when at least n_pop ^1^/2 elements having maximum coverage were obtained	36	n/a	9	13	n/a	n/a	n/a	n/a	34	22
the number of stimulus sets that achieve maximum coverage	24 (1)	2 (2)	172 (2)	125 (1)	3 (3)	1 (1)	2 (1)	3 (3)	12 (12)	111 (1)

^1^ n_pop represents the initial number of individuals per population and is equal to 20 in current implementations.

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
