# Peer review of "Cost-Efficient Approaches for Fulfillment of Functional Coverage during Verification of Digital Designs"

_micromachines, 2022, doi:10.3390/mi13050691_

Round 1
Reviewer 1 Report
Digital integrated circuits play important role in development new information technologies and support the Industry 4.0 from the hardware point of view. An increasing both structural and functional complexity of the state-of-the-art digital ICs provides the rise of a design time and particularly a time cost on verification. Development of automated techniques and tools for the effort simplification on verification and debugging the complex digital projects is challenge and very actual topic in the area of microelectronic design.
There are several professional verification EDA tools from the world leading vendors of CAD tools. But all these systems are essentially expansive and not always available for small design centers and academia. Therefore, development of efficient open source or low budget verification tools for digital ICs can be interesting for mentioned above potential customers.
The task of verification is the NP-hard problem due to high complexity of up-to-date digital IC. Some heuristics based on machine learning, genetic algorithms, bio-inspired approaches, etc. are proposed for a reducing complexity of this task. Meanwhile, the tradeoff between minimum time on functional verification and maximum coverage is the key point of all proposed techniques.
The paper under review aims to automate the functional verification process by generating relevant input stimuli to the thorough simulation and assessment of the digital ICs behavior. The author’s team considers some possibilities for the functional verification of digital circuits empowered with the usage of genetic algorithms. Unfortunately, the Abstract and Introduction section do not contain the clear description of goals, object(s) and subject(s) for the proposed paper and research work being carried out. The completeness of description, correctness of conclusions and discussions, and the attainment of the goals cannot be estimated without properly organized problem statement.
Authors have revised the previous version of proposed paper and reflected some of the principal comments after the first reviewing.
Meanwhile, the following comments are not considered in the new version of the paper:
- The genetic algorithms as a variant of evolutionary computation in optimization problem should be described according to the math statement. Current version of paper does not contain a mapping of the object to the genetic algorithm terms, a description of gene or chromosome structure and fitness function.
- The flow diagram in Fig. 2 demonstrates some formal sequence of operations, some of which are complex and principal problems, but there are no detailed descriptions for them in text. Authors have improved a description of the flow diagram, but interaction with ModelSim is not represented again.
- The computational complexity for the proposed solutions did not estimate as well as the time cost.
The current version is look like better, but essentially increased in number of pages. Authors use an exhaustive text explanation even in cases when can be used tabular form, for instance at consideration of relevant works.
Reviewer 2 Report
The topic of the paper and the result is interesting. However, there are some points to be clarified to be accepted as a publication. Please kindly consider the following suggestions.
- Please mention in a quantitative manner, how much the proposed method performance outperforms the existing method in the abstract
- The presentation of figure 7-10 is quite confusing. For a group of figures with one caption, it is better to be written on the same page. If it is separated into two pages, it is better to write the caption on each page. For 1 caption with many figures, please also add a,b, and c to explain each figure in that caption. The axis label is also difficult to read.
- There are so many figures in Figures 7-10 but the explanation is not enough. Please discuss the result for these figures more in a quantitative manner. I believe there are so many interesting points to be discussed from these figures.
- Overall, the results presented in tables or figures are quite confusing. It is not clear which one is the existing method or the proposed method. Please clearly identify each method. If it is a proposed method, please write (proposed). If it is an existing method, please cite the reference to identify it.
Author Response
Please see the attachment.

This manuscript is a resubmission of an earlier submission. The following is a list of the peer review reports and author responses from that submission.
Round 1
Reviewer 1 Report
The authors propose the use of a genetic algorithm for the automated creation of input patterns used for the functional verification of digital designs. The contributions the authors aim at include the implementation of the system, the design of the evolutionary operators used for the implementation of the genetic algorithm, and the execution and discussion of experiments to evaluate the proposed approach.
The paper contains little to no novelty, since the (obvious) idea of using genetic algorithms instead of randomly generating test patterns has been proposed in previous papers. While the authors claim to be superior to existing approaches (needing fewer evaluations than SL based approaches and relying on more expressive Objective models than existing GA-based methods), these claims are neither discussed nor experimentally evaluated.
The proposed implementation of the genetic algorithm is primitive and quite bizarre since it ignores many practices which are widely accepted in the optimization community, such as a randomized selection and mating scheme or an archive-based implementation of elitism. The dynamic change of the mutation rate, which the authors mention as one of the contributions of the paper, appears particularly odd, since it increases the probability of mutation towards the late stages of the optimization, violating the practice of starting with a high exploration rate (more mutations) and then shifting towards the exploitation (fewer mutations) as the optimization goes on. Overall, my guess would be that any of the established genetic algorithms (e.g., NSGA2 or SPEA), all of which come with freely available high-quality implementations, would significantly outperform the presented approach (a comparison to existing SOTA algorithms should be included in the paper anyway to justify the need for custom-made evolutionary operators).
In summary, the paper does not provide any novel observations or contributions (since it lacks experimental comparisons to both the SOTA optimization approaches and to existing approaches in the domain of verification automation). As a minor note: Particularly the first part of the manuscript contains a large number of grammatical errors and requires substantial proofreading.
Reviewer 2 Report
Digital integrated circuits play important role in development new information technologies and support the Industry 4.0 from the hardware point of view. An increasing both structural and functional complexity of the state-of-the-art digital ICs provides the rise of a design time and particularly a time cost on verification. Development of automated techniques and tools for the effort simplification on verification and debugging the complex digital projects is challenge and very actual topic in the area of microelectronic design.
There are several professional verification EDA tools from the world leading vendors of CAD tools. But all these systems are essentially expansive and not always available for small design centers and academia. Therefore, development of efficient open source or low budget verification tools for digital ICs can be interesting for mentioned above potential customers.
The task of verification is the NP-hard problem due to high complexity of up-to-date digital IC. Some heuristics based on machine learning, genetic algorithms, bio-inspired approaches, etc. are proposed for a reducing complexity of this task. Meanwhile, the tradeoff between minimum time on functional verification and maximum coverage is the key point of all proposed techniques.
The paper under review aims to automate the functional verification process by generating relevant input stimuli to the thorough simulation and assessment of the digital ICs behavior. The author’s team considers some possibilities for the functional verification of digital circuits empowered with the usage of genetic algorithms. Unfortunately, the Abstract and Introduction section do not contain the clear description of goals, object(s) and subject(s) for the proposed paper and research work being carried out. The completeness of description, correctness of conclusions and discussions, and the attainment of the goals cannot be estimated without properly organized problem statement.
There is the following list of the principal comments to the proposed paper:
- Authors has postulated the “digital designs” as a general type of devices under consideration, but then has reduced the target set to FPGA devices. The type of circuit has essential influence on the design flow and therefore very important to be specified for a design automation purpose.
- The genetic algorithms as a variant of evolutionary computation in optimization problem should be described according to the math statement. Current version of paper does not contain a mapping of the object to the genetic algorithm terms, a description of gene or chromosome structure and fitness function.
- The flow diagram in Fig. 2 demonstrates some formal sequence of operations, some of which are complex and principal problems, but there are no detailed descriptions for them in text. For instance, Step 2 “Evaluation the parents in population using ModelSim simulations” does not explain the set of estimated parameters or characteristics, the aim of evaluation, the interaction with Siemens EDA, etc. Step 3 “Find and save on disk the best solution available” does not represent the criteria for the best solution, the proof that this solution is “the best” really, how such “find” is implemented.
- The computational complexity for the proposed solutions did not estimate as well as the time cost.
- The correct name of the ModelSim simulator’s vendor is Mentor Graphics (not just a Mentor), and Siemens EDA is a successor of Mentor Graphics nowadays.
- The draft of eco-system depicted on Fig. 3 is not universal because doesn’t rely on the configuration for different digital designs and doesn’t reflect the roles of verification engineer and design engineer as well as requirements to their qualification.
- That is the reason of combination Verilog and SystemVerilog languages?
- The object for experimental part is questionable. A verbal description of behavior for the train electronic management system is incomplete for understanding. Why the standard benchmark circuit(s) for the functional verification case studies did not use under experimental research?
- The experimental results are dealt with the set of proposed algorithms only without consideration alternative solutions and comparative analysis with results of other authors.
- Authors declared the time reduction at functional verification as the effect of genetic algorithms application, but experimental results don’t reflect this characteristic at all.
- There are not even any recommendation what algorithm among considered may be used at automation the procedure of functional verification in different cases and conditions, how the eco-system can select the proper algorithm in real practice.
- The use of self-cited references [1]-[4] in the first paragraph of Introduction section is not correct, because declared general aspects are confirmed by the individual papers of the same authors, but not the different research groups.
- At the first glance, the paper is look like the engineering report, but not the structured research paper. There is fuzzy relationship between title, actual proposed solution, case study and conclusion.
The reviewer's decision is the following under the aggregation of the mentioned above comments. The paper cannot be accepted for publication in the high-ranking journal as “Micromachines” without essential revision and improvement.